# Benefits of Patient-Specific Reconstruction Plates in Mandibular Reconstruction Surgical Simulation and Resident Education

**DOI:** 10.3390/jcm11185306

**Published:** 2022-09-09

**Authors:** Khanh Linh Tran, Matthew Lee Mong, James Scott Durham, Eitan Prisman

**Affiliations:** Division of Otolaryngology, Department of Surgery, Faculty of Medicine, University of British Columbia, Vancouver, BC V6T 1Z4, Canada

**Keywords:** virtual surgical planning, CAD/CAM, mandibular reconstruction, patient-specific reconstruction plates, medical education, surgical simulation

## Abstract

Poorly contoured mandibular reconstruction plates are associated with postoperative complications. Recently, a technique emerged whereby preoperative patient-specific reconstructive plates (PSRP) are developed in the hopes of eliminating errors in the plate-bending process. This study’s objective is to determine if reconstructions performed with PSRP are more accurate than manually contoured plates. Ten Otolaryngology residents each performed two ex vivo mandibular reconstructions, first using a PSRP followed by a manually contoured plate. Reconstruction time, CT scans, and accuracy measurements were collected. Paired Student’s *t*-test was performed. There was a significant difference between reconstructions with PSRP and manually contoured plates in: plate-mandible distance (0.39 ± 0.21 vs. 0.75 ± 0.31 mm, *p* = 0.0128), inter-fibular segment gap (0.90 ± 0.32 vs. 2.24 ± 1.03 mm, *p* = 0.0095), mandible-fibula gap (1.02 ± 0.39 vs. 2.87 ± 2.38 mm, *p* = 0.0260), average reconstruction deviation (1.11 ± 0.32 vs. 1.67 ± 0.47 mm, *p* = 0.0228), mandibular angle width difference (5.13 ± 4.32 vs. 11.79 ± 4.27 mm, *p* = 0.0221), and reconstruction time (16.67 ± 4.18 vs. 33.78 ± 8.45 min, *p* = 0.0006). Lower plate-mandible distance has been demonstrated to correlate with decreased plate extrusion rates. Similarly, improved bony apposition promotes bony union. PSRP appears to provide a more accurate scaffold to guide the surgeons in assembling donor bone segments, which could potentially improve patient outcome and reduce surgical time. Additionally, in-house PSRP can serve as a low-cost surgical simulation tool for resident education.

## 1. Introduction

The human mandible is an important anatomical structure that functions in respiration, deglutition, and mastication [1]. Conditions that affect the mandible, such as oral cancer, osteoradionecrosis, or osteomyelitis, can affect patients’ oral functioning and quality of life [2]. Mandibular resection and subsequent reconstruction with donor fibular free flap is a technique to manage oral disease and restore aesthetic and functional outcomes for head and neck cancer patients. The surgery involves resecting the diseased portion of the mandible, harvesting the fibular flap, contouring the fibula segments, securing the flap to the mandible, and vascular anastomosis [3]. To secure the fibular free flap segments to the non-resected area of the mandible, a titanium plate is typically employed [2]. In the traditional fashion, in order to faithfully restore the mandible contour, the plate is bent intraoperatively to the native mandible. It serves as a template for the shaping and placement of the fibula reconstruction. This is a time-consuming step and could take up to 60 min in the operating room [4]. Consequences of a poorly bent plate include plate fracture, plate exposure, and malunion/nonunion of the reconstruction [5]. These complications can severely affect patients’ quality of life and may involve costly and invasive treatment such as antibiotics or additional surgeries [6,7,8].

Alternatively, customized titanium reconstruction plates for a patient-specific reconstruction (patient-specific reconstruction plate, or PSRP) can be designed, optimized for overall strength, and manufactured using selective layer melting or computer numerical control milling [9]. These customized plates and cutting guides are often provided by a commercial medical device company and are associated with incremental costs of up to 3000–8200 USD per case [10,11]. Through a comparative case series, Sieira-Gil et al. demonstrate that reconstructions using VSP with custom titanium plates result in better dental occlusion, lower plate exposure rate, and lower operative time compared with traditional, un-guided surgeries without custom plates [12]. Wilde et al. determined through a multicenter clinical study that the cost associated with plate manufacture can be offset by the time saved in the operating room [13]. However, the specific effect of PRSP versus traditional manual contouring of reconstruction plates for VSP in reducing the gap distance between fibula segment as well as the distance between the plate and reconstruction, which are associated with nonunion and plate exposure, respectively, has not been clearly demonstrated. Furthermore, the ability to generate in-house PSRP has, to the authors’ knowledge, never been demonstrated.

The purpose of this study is three-fold: (1) to demonstrate a proof-of-concept design for PSRP using in-house VSP software, (2) to compare PSRP with manually contoured plates in mandibular reconstruction, and (3) to report on the use of customized plates in resident education.

## 2. Materials and Methods

This study was approved by the clinical research ethics board at the author’s institution (H21-00205). Surgical trainees in the Division of Otolaryngology at a tertiary care center were recruited and consented to the study. The participants performed a simulation of mandibular reconstruction with the fibular bone. The mandibular defect chosen involves the mandible ramus and body and crosses the midline to the contralateral side. The in-house VSP software developed at the authors’ institution was utilized to plan for a three-piece reconstruction [14]. In order to isolate the specific effect of the preprinted PSRP on the reconstruction, the two remaining segments of the mandible and the three pieces of the fibula and the reconstruction model were 3D-printed in acrylonitrile butadiene styrene. The participants first viewed a demonstration by the senior authors, then performed two reconstructions: first with a 2.0 mm patient-specific, pre-printed vinyl plate (PSRP) and then with a standard, straight 2.0 mm titanium plate that requires manual contouring to the printed reconstruction model. (Stryker, reference number 55-15719). To avoid confounding results, reconstructions with PSRP were performed first to limit any potential advantages over manually contoured plates. In the PSRP component, participants performed the reconstructions in the following steps: drilled holes on the mandible corresponding to the PSRP, arranged the fibula segments, and secured the mandible and fibula segments using the reconstruction plate. In the manual contouring group, this process was preceded by manually contouring the titanium plate to the reconstruction model.

Time to perform each step of the reconstruction was recorded. CT scans of the participants’ reconstruction were obtained and segmented using a 3D Slicer, where the following measurements were also performed: volumetric overlap and Hausdorff-95 distance of the reconstructive segments and the entire mandible [15]. Volumetric overlap is calculated as twice the volume of the intersection between the actual and planned reconstructions divided by the sum of the volumes between the two models [15]. Hausdorff-95 is the 95th percentile of the Hausdorff distance, which is the maximum distance of the minimum distances between each vertex on the actual and planned reconstructions [15]. In CloudCompare, a 3D mesh model processing platform that supports computations such as the distance between different models, the distances between the reconstructive plate and mandible, between each fibula segment, and between the mandible and fibula segments were obtained. Each measurement is illustrated in Figure 1. Statistical analyses were performed in Microsoft Excel. A paired Student’s *t*-test was performed to compare the measurements between the reconstruction groups, with each participant acting as their own control. A *p*-value of <0.05 was considered significant.

## 3. Results

In total, ten Otolaryngology residents in training, two from each year one to year five of training, consented and participated in this study. Six male and four female participants were included in the study. Junior residents are those who are in years one to three of training, and senior residents are in years four and five. They have observed, on average, 18.8 ± 7.0 free flap reconstruction surgeries and have assisted in 11.7 ± 5.5 cases.

### 3.1. Reconstruction Time

The total time to assemble the reconstruction and time to contour the plate in the two groups is recorded in Table 1. There is a significant decrease in reconstructive time in the PSRP group for all participants and junior residents, as well as senior residents, to a non-significant degree.

### 3.2. Reconstruction Accuracy

Lower HD-95 distance of the fibula segments, the lower distance between reconstructive plate and mandible, the lower gap distance between each fibula segment and between the mandible and fibula segments, and lower mandible angle width deviation were observed in the 3D-printed plate group to a significant degree (Table 2). There were also non-significant improvements in volumetric overlap, HD-95 for the reconstructed mandible, and difference in mandible ramus-condyle height.

## 4. Discussion

Mandible reconstruction is a challenging procedure, even for experienced surgeons. Complications of the surgery range from reconstructive plate extrusion, non-union or malunion of the flap segments, to free flap failure, all with devastating consequences to patients’ cosmetic, oral functioning, and quality of life [5]. Virtual surgical planning has been successfully applied to improve these outcomes. However, they remain a challenge. At the authors’ center, an in-house VSP software has been developed and used for oromaxillofacial surgical reconstruction since 2017 [14,16,17,18,19,20]. The platform has been adapted to allow for patient-specific reconstruction plate modeling. To the authors’ knowledge, there has not been an evaluation of the utility of in-house designed reconstruction plates in the context of resident training. As such, this study investigates the potential of applying an open-source, in-house VSP platform to generate PSRP by comparing mandible reconstructions with PSRP versus manually contoured plates.

Inaccurate plate contouring could result in additional tensional and torsional forces being placed on the temporomandibular (TMJ) joint, leading to malocclusion or injury to the TMJ [21,22,23]. Malocclusion could severely impact patients’ quality of life and may require additional surgery to remove the plate and realign the position of the reconstruction [24]. The results of this study, including higher volumetric overlap (*p* = 0.080), lower Hausdorff-95 distance (*p* = 0.038), and lower difference in mandible angle width (*p* = 0.003), suggest that employing PSRP can potentially improve reconstructive accuracy as compared with manually bent plates. Manual contouring of reconstructive plates can also introduce residual stress to the plate, leading to postoperative plate fracturing [25]. In a series of biomechanical tests, Gutwald et al. have shown that preprinted plates present an opportunity for customization to reduce stress [26]. Overall, preprinted plates can withstand stronger force than manually bent plates as they avoid deformity introduced during the contouring process [26].

Plate exposure is a common hardware-related complication in head and neck reconstruction with a free vascularized flap. The rate of plate exposure is reported to be between 9–20% [6,7,8]. Management of plate exposure ranges from antibiotics to hardware removal and surgical debridement of the flap. Treatment not only affects patients’ long-term quality of life but can also present a significant cost to patients and the healthcare system [8]. Although flap selection and radiation therapy contribute to the rate of plate extrusion, plate-to-bone gap distance is hypothesized to be an important factor [7,27,28]. Chepeha et al. suggest that the dead space between the plate and bone induces the contraction of tissue lateral to the plate to fill in the void, resulting in eventual flap necrosis [29]. Furthermore, in a clinical study of 94 patients who underwent mandibular reconstruction, Davies et al. demonstrate an association between lower plate-to-bone gap distance and lower rates of plate exposure and intraoral dehiscence [30]. In our study, patient-specific reconstruction plates are shown to reduce the gap between the mandible and plate compared with manually contoured plates. Since the study controls for variability in mandible defect and fibula segment sizes, the difference in the two groups is likely attributable to the type of plate used. Although further clinical studies are required to confirm the effect of printed plates in lowering plate-to-bone gap distance in vivo, the results of this study show a promising utility in designing and manufacturing PSRP.

Reconstruction plates secure the mandible and reconstruction segments to each other to promote union [31]. If the bone does not fully heal, the segments may either not fuse or inadequately fuse together, resulting in nonunion or malunion, respectively [32]. Incomplete osseointegration is associated with a higher rate of wound complications and could lead to bone fracture, osteonecrosis, or reabsorption. If malunion or nonunion is detected, patients may undergo surgical plate refixation or a secondary bone graft [6]. The rate of nonunion is reported to be between 5–23% for mandible reconstruction [6,7,32,33,34,35,36,37]. Swenseid et al. demonstrate that a gap distance of 1 mm or greater significantly increases the likelihood of malunion or nonunion developing [32]. In the present study, the reconstructions with hand-bent plates resulted in a gap of over 2 mm, on average. Significantly, both the gap distances between mandible-fibula and fibula-fibula in the hand-bent group are more than twice as large as that of the group with PSRP. In the former group, errors introduced by inaccurate manual contouring of the reconstruction plate propagated in later stages of assembling and securing the fibula segments, leading to significant gaps in the final reconstruction. Although our study is ex vivo by nature, the average gap distance measured suggests that PSRP have the potential to improve the osseointegration rate of mandible reconstructions.

The time taken to perform a reconstruction is twice as long for the group with manually contoured plates compared with the group with PSRP. This result represents potential cost savings in the operating room as well as a reduction in complications, which has been shown to be associated with lower operation time [38]. Simulation has been increasingly used as a training tool for otolaryngology surgical residents [21,39]. VSP represents an opportunity for residents to learn and practice complicated procedures such as mandible reconstructions in a controlled environment outside the operating room. The high cost of surgical titanium plates, estimated at 800 CAD per plate, prevents them from being used for routine training purposes. Therefore, in the study, vinyl was explored as a cheaper alternative for this particular reconstruction simulation that does not require the longevity and biocompatibility of in vivo use. Moreover, the reconstruction time is reduced for both junior and senior residents when using PSRP compared with manually contoured plates. Thus, this result suggests that PSRP can benefit residents with differing experience levels.

There are several limitations to the study. Since the participants are surgical trainees, the benefits of using 3D-printed PSRP with respect to time-saving and higher accuracy may not be replicated in more experienced head and reconstructive neck surgeons. However, 3D-printed PSRP may still be used as an educational and training tool in mandibular reconstruction surgical simulation by lowering the learning curve associated with plate contouring. Another limitation is that the 3D-printed PSRP is manufactured with vinyl as opposed to titanium. Although further in vivo studies with patient-specific titanium plates need to be carried out to definitively compare their benefits with manually contoured plates, vinyl plates could be used as a low-cost training tool alternative to titanium plates.

## 5. Conclusions

Customized mandibular reconstruction plates can be readily designed with open source, in-house software. These customized reconstruction plates improve the accuracy of executing VSP for mandible reconstruction compared with manually contoured mandible reconstruction plates for less experienced surgeons. Finally, this in-house design can be used in mandible reconstruction training as a low-cost simulation tool. Potential clinical benefits of utilizing patient-specific reconstruction plates include surgical time reduction, improved reconstructive accuracy, and lower complication rates, which warrants further clinical studies.

## Figures and Tables

**Figure 1 jcm-11-05306-f001:**
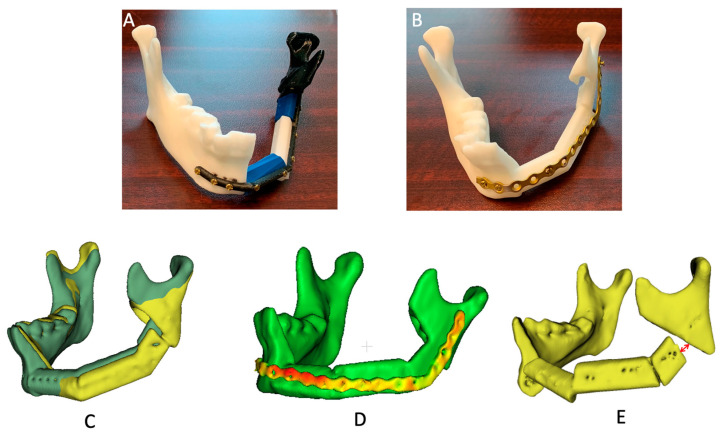
Reconstructions and structural accuracy variables. (**A**) Reconstruction with a customized, patient-specific, pre-printed plate. (**B**) Reconstruction with a manually contoured plate. (**C**) Volumetric overlap and Hausdorff-95 between reconstruction (yellow) and plan (green) were measured in 3D Slicer. (**D**) Distance between reconstruction plate and mandible was measured in CloudCompare. (**E**) Distance at each apposition was measured in 3D Slicer (red arrow).

**Table 1 jcm-11-05306-t001:** Comparison of time to perform reconstructions in each group. Significant *p*-values are indicated with a (*).

	Manual Contour Plate (*n* = 10)	Patient-Specific, Reconstruction Plate (PSRP) (*n* = 10)	*p*-Value
Time to perform reconstruction (min)	33.78 ± 8.45	16.67 ± 4.18	<0.001 *
(a) Junior residents (min)	39.4 ± 3.78	17.8 ± 3.35	<0.001 *
(b) Senior residents (min)	29.25 ± 8.54	15.25 ± 5.19	0.119
Time to contour plate (min)	13.33 ± 3.97	N/A (Not applicable)	
(a) Junior residents (min)	15.2 ± 2.49	N/A	
(b) Senior residents (min)	11.0 ± 4.55	N/A	

**Table 2 jcm-11-05306-t002:** Average accuracy measurements of the reconstructions in each group. Significant *p*-values are indicated with a (*).

	Manual Contour Plate Plate (*n* = 10)	PSRP Plate (*n* = 10)	*p*-Value
Volumetric overlap of reconstructed mandible (%)	65.70 ± 11.33	70.40 ± 8.62	0.310
Volumetric overlap of fibula segments (%)	51.70 ± 14.61	63.00 ± 12.56	0.080
Hausdorff-95 of reconstructed mandible (mm)	2.40 ± 1.14	1.75 ± 0.90	0.175
Hausdorff-95 of fibula segments (mm)	3.41 ± 1.47	2.10 ± 1.12	0.038 *
Plate-mandible gap distance (mm)	0.75 ± 0.31	0.39 ± 0.21	0.007 *
Gap distance between fibula segments (mm)	2.24 ± 1.03	0.90 ± 0.32	0.001 *
Gap distance between mandible and fibula segments (mm)	2.87 ± 2.38	1.02 ± 0.39	0.026 *
Difference in mandible angle width (mm)	11.79 ± 4.27	5.13 ± 4.32	0.003 *
Difference in mandible ramus-condyle height (mm)	1.26 ± 1.52	1.19 ± 0.93	0.902

## Data Availability

Data sharing is not applicable.

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
