# Peer review of "Benefits of Patient-Specific Reconstruction Plates in Mandibular Reconstruction Surgical Simulation and Resident Education"

_jcm, 2022, doi:10.3390/jcm11185306_

Round 1

Reviewer 1 Report

In the surgeries performed for mandible reconstruction with an experimental method created ex-vivo in their study, the authors; With the virtual surgery planning platform developed in-house; They compared the manually tilted plates. They also planned to investigate the effect of this planning on surgical time and the contribution of this platform to resident training. And as a result of their studies, they stated that pre-printed plates gave better results than manually tilted plates.

If we look at the article as a whole, it appears as a well-designed study. However, as of the subject of the article, it has been studied in previous years and is currently used on patients. Again, as stated in the sources of the authors, the use of surgical simulation tools in previous years contributed to the training of residents. In this context, the article does not shed a new perspective and light on the literature.

Title: The benefits of patient-specific mandible reconstruction plates have been demonstrated in previous studies. For this reason, a different title can be searched rather than using a previously known subject with a question sentence in the title.

Abstract: ok

Introduction: This section is way too long. A large amount of literature information is included. Only 15 references are cited in this section. In fact, what should be done is to give a brief information about patient-specific reconstructive plaques in the literature, and then write what they plan to reveal in their own studies, unlike the literature.

Methods: The terms Hausdorff-95 and Cloud Compare mentioned in this section are, I think, mathematical expressions. I think it would be appropriate to give a brief information about them.

Results: In this section, it is said that 10 otolaryngology assistants participated in the study. But then there are 6 male and 5 female participants. It would be appropriate to correct this inconsistency.

Discussion:ok

Conclusions:ok

References: I think that the number of sources on the basis of this article is quite high. I think it would be better to reduce it if possible.

Kind regards

Author Response

Response to reviewers

We would like to thank the reviewers for their time in reviewing the manuscript and their insightful comments. We have taken each comment into careful consideration and revised the manuscript as follows.

Reviewer 1

In the surgeries performed for mandible reconstruction with an experimental method created ex-vivo in their study, the authors; With the virtual surgery planning platform developed in-house; They compared the manually tilted plates. They also planned to investigate the effect of this planning on surgical time and the contribution of this platform to resident training. And as a result of their studies, they stated that pre-printed plates gave better results than manually tilted plates.

If we look at the article as a whole, it appears as a well-designed study. However, as of the subject of the article, it has been studied in previous years and is currently used on patients. Again, as stated in the sources of the authors, the use of surgical simulation tools in previous years contributed to the training of residents. In this context, the article does not shed a new perspective and light on the literature.

Title: The benefits of patient-specific mandible reconstruction plates have been demonstrated in previous studies. For this reason, a different title can be searched rather than using a previously known subject with a question sentence in the title.

  • Response: We have revised the title to reflect the benefits of PSRP in the context of surgical simulation and resident education.

“Benefits of Patient-Specific Reconstruction Plates in Mandibular Reconstruction Surgical Simulation and Resident Education”

Abstract: ok

Introduction: This section is way too long. A large amount of literature information is included. Only 15 references are cited in this section. In fact, what should be done is to give a brief information about patient-specific reconstructive plaques in the literature, and then write what they plan to reveal in their own studies, unlike the literature.

  • Response: We have removed a portion of the Introduction pertaining to virtual surgical planning which we discussed in other sections. We have kept the remainder of the Introduction as it gives the readers an overview of the process of mandibular reconstruction and introduces them to PSRP.

Methods: The terms Hausdorff-95 and Cloud Compare mentioned in this section are, I think, mathematical expressions. I think it would be appropriate to give a brief information about them.

  • Response: We have added an explanation of the volumetric overlap and Hausdorff-95 metrics to paragraph 2 of the Methods section.

“Volumetric overlap is calculated as twice the volume of the intersection between the actual and planned reconstructions divided by the sum of the volumes between the two models.15 Hausdorff-95 the 95th percentile of the Hausdorff distance, which is the maximum distance of the minimum distances between each vertex on the actual and planned reconstructions.15

As well, we added an explanation of what CloudCompare is to paragraph 2 of the Methods sections.

“In CloudCompare, a 3D mesh model processing platform that supports computations such as the distance between different models...”

Results: In this section, it is said that 10 otolaryngology assistants participated in the study. But then there are 6 male and 5 female participants. It would be appropriate to correct this inconsistency.

  • Response: Thank you for noting this discrepancy. We have corrected the number of participants to “6 male and 4 female”.

Discussion:ok

Conclusions:ok

References: I think that the number of sources on the basis of this article is quite high. I think it would be better to reduce it if possible.

  • Response: By removing a part of the Introduction and Discussion, we have removed 5 sources from the References section.

Reviewer 2 Report

The manuscript title (What are the benefits of Patient-Specific Mandibular Reconstruction Plates: An Ex Vivo Simulation Study). This study’s objective is to determine if reconstructions performed with PSRP are more accurate compared to manually contoured plates.

Materials and Methods:

- Is there any standardization or calibration session?

- CloudCompare, would you explain the process and the accuracy?

- " between the reconstructive plate and mandible, between each fibular segment, and between the mandible and fibular segments were obtained" why specifically were these locations where measured and compared?

- Would you explain more about the control group?

Results:

-" In total, ten ...." However, later you mentioned "Six male and five female"?

- "Junior and Senior resident" You mentioned two from each year, would you clarify?

Discussion:

- The first few paragraphs are repeated from the introduction, please revise.

- "Overall, preprinted plates can withstand stronger force than manually bent plates.." Source?

-  The cost-effectiveness was repeated in multiple areas in the discussion, please revise.   

Best

Author Response

Response to reviewers

We would like to thank the reviewers for their time in reviewing the manuscript and their insightful comments. We have taken each comment into careful consideration and revised the manuscript as follows.

Reviewer 2

The manuscript title (What are the benefits of Patient-Specific Mandibular Reconstruction Plates: An Ex Vivo Simulation Study). This study’s objective is to determine if reconstructions performed with PSRP are more accurate compared to manually contoured plates.

Materials and Methods:

- Is there any standardization or calibration session?

  • Response: Although we did not hold a standardization/calibration session prior to the plating course, there were efforts to control for variables during the course. In order to control for the participants’ varying experience with performing mandibular reconstruction, we 3D-printed the mandible and fibula segments instead of asking participants to perform the simulated mandible and fibula resections. In addition, the senior authors (Dr. Prisman and Dr. Durham) who are experienced head and neck surgeons, demonstrated to the participants how to perform each reconstruction. Also, to avoid the participants performing better on the second reconstruction in terms of shortened time and improved accuracy due to familiarity with the process from the first reconstruction, we arranged the order of the reconstructions such that the one with PSRP was carried out first. This is mentioned in paragraph 1 of the Materials and Methods section.

“In order to isolate the specific effect of the preprinted PSRP on the reconstruction, the two remaining segments of the mandible and the three pieces of the fibula, and the reconstruction model were 3D-printed in acrylonitrile butadiene styrene. The participants first viewed a demonstration by the senior authors, then performed two reconstructions: first with a 2.0 mm patient-specific, pre-printed vinyl plate and then with a standard, straight 2.0 mm titanium plate that require manual contouring to the printed reconstruction model. (Stryker, reference number 55-15719). To avoid confounding results, reconstructions with PSRP were performed first to limit any potential advantages over manually contoured plates.”

- CloudCompare, would you explain the process and the accuracy?

  • Response: We added an explanation of what CloudCompare is to paragraph 2 of the Methods sections.

“In CloudCompare, a 3D mesh model processing platform that supports computations such as the distance between different models...”

- " between the reconstructive plate and mandible, between each fibular segment, and between the mandible and fibular segments were obtained" why specifically were these locations where measured and compared?

  • Response: All of the distances listed are measured to evaluate the quality of the reconstructive. In an ideal reconstruction, those distances are minimized to optimize for lower plate extrusion (distance between reconstructive plate and mandible) and higher rates of bony union (distances at each apposition where the fibular segments or fibular segments and native mandible come into contact).

- Would you explain more about the control group?

  • Response: The control group is the second reconstruction that was performed wherein the plate was manually contoured to the 3D-printed mandible reconstruction model as opposed to patient-specific reconstruction plate. This is mentioned in the first paragraph of the Introduction:

“In the traditional fashion, in order to faithfully restore the mandible contour, the plate is bent intraoperatively to the native mandible and serves as a template for the shaping and placement of the fibula reconstruction.”

And the first paragraph of the Materials and Methods section:

“The participants first viewed a demonstration by the senior authors, then performed two reconstructions: first with a 2.0 mm patient-specific, pre-printed vinyl plate and then with a standard, straight 2.0 mm titanium plate that require manual contouring to the printed reconstruction model.”

Results:

-" In total, ten ...." However, later you mentioned "Six male and five female"?

  • Response: Thank you for noting this discrepancy. We have corrected the number of participants to “6 male and 4 female”.

- "Junior and Senior resident" You mentioned two from each year, would you clarify?

  • Response: We have added the explanation for this classification.

“Junior residents are those who are in year one to three of training, and senior residents are in year four and five.”

Discussion:

- The first few paragraphs are repeated from the introduction, please revise.

  • Response: We have removed the part of the Introduction on virtual surgical planning, which is mentioned in the Discussion paragraph 1. As well, we have removed the introduction to PSRP from the Discussion paragraph 1 as it overlaps with information presented in the Introduction.

- "Overall, preprinted plates can withstand stronger force than manually bent plates.." Source?

  • Response: We added the reference to Gutwald et al. (2017) which was referenced in the preceding sentence.

“In a series of biomechanical tests, Gutwald et al. has shown that preprinted plates present an opportunity for customization to reduce stress.26 Overall, preprinted plates can withstand stronger force than manually bent plates as they avoid deformity introduced during the contouring process.26

-  The cost-effectiveness was repeated in multiple areas in the discussion, please revise.  

  • Response: The potential cost-savings benefits of PSRP are due to several factors, including:
    • Reduction in costs of treatment to manage complications (Discussion, paragraph 3):

“Treatment not only affects patients’ long-term quality of life, but can also present a significant cost to patients and the healthcare system.”

  • Reduction in costs associated with lower OR time (Discussion, paragraph 5):

“This result represents potential cost savings in the operating room as well as reduction in complications, which has been shown to be associated with lower operation time.”

As well, exploring the option of in-house designed and manufactured PSRP could help address the cost-prohibitive barrier of commercial solutions (Discussion, paragraph 1):

“The cost-prohibitive nature of commercial solutions, which have been reported to be as high as 8,000 USD per case, may render them inaccessible to most surgical centers and patients.”

However, since the cost of manufacturing a titanium plate can be quite high, vinyl was explored as an alternative option (Discussion, paragraph 5):

“The high cost of surgical titanium plates, estimated at 800 CAD per plate, prevents them from being used for routine training purposes.”

As the cost-savings potential of PSRP in general are not limited to the in-house solution that we explored in this study and are closely related to each of the benefit that PSRP can provide, we have left the cost-effectiveness discussion in the original locations. In addition, the cost-prohibitive aspect of commercial solutions serves as an important introduction in the Discussion (paragraph 1) as to why we explored in-house solutions at our institution, whereas using vinyl as an alternative to titanium pertains to the educational value of the plates we used in this study (paragraph 5). Thus, we have also left those cost-benefits consideration in the original locations.

Reviewer 3 Report

Title: What are the benefits of Patient-Specific Mandibular Reconstruction Plates: An Ex Vivo Simulation Study

Journal: Journal of Clinical Medicine

This manuscript was intended to determine if mandibular reconstructions performed with PSRP are more accurate compared to manually contoured plates. This ex-vivo study used a simulation of mandibular reconstruction with the fibular bone.

This is a concise and well written article.

Only one remark:

-          The abbreviation ‘VSP’ should be explained at the first usage in the text including the abstract. Hence, the sentence ’Additionally, in-house VSP can a low-cost surgical simulation tool for resident education.’ should be corrected.

Author Response

Response to reviewers

We would like to thank the reviewers for their time in reviewing the manuscript and their insightful comments. We have taken each comment into careful consideration and revised the manuscript as follows.

Reviewer 3

Title: What are the benefits of Patient-Specific Mandibular Reconstruction Plates: An Ex Vivo Simulation Study

Journal: Journal of Clinical Medicine

This manuscript was intended to determine if mandibular reconstructions performed with PSRP are more accurate compared to manually contoured plates. This ex-vivo study used a simulation of mandibular reconstruction with the fibular bone.

This is a concise and well written article.

Only one remark:

-          The abbreviation ‘VSP’ should be explained at the first usage in the text including the abstract. Hence, the sentence ’Additionally, in-house VSP can a low-cost surgical simulation tool for resident education.’ should be corrected.

  • Response: We have revised that sentence in the Abstract paragraph 4 to:

“Additionally, in-house PSRP can serve as a low-cost surgical simulation tool for resident education.”

Round 2

Reviewer 1 Report

When I examined the revision version of the article, I saw that the authors made the desired changes and fixed some errors. However, as I mentioned in my first review, the subject of the article has been discussed in the literature before in clinical studies. In this respect, I believe that it will not make an additional contribution to the literature.

Kind regards